# Interaction of Biochar Addition and Nitrogen Fertilizers on Wheat Production and Nutrient Status in Soil Profiles

**DOI:** 10.3390/plants13050614

**Published:** 2024-02-23

**Authors:** Jiale Liu, Zirui Chen, Si Wu, Haijun Sun, Jincheng Xing, Zhenhua Zhang

**Affiliations:** 1Co-Innovation Center for Sustainable Forestry in Southern China, College of Soil and Water Conservation, Nanjing Forestry University, Nanjing 210037, China; liujiale@njfu.edu.cn (J.L.); chenzr@njfu.edu.cn (Z.C.); nlwus@njfu.edu.cn (S.W.); 2Institute of Jiangsu Coastal Agricultural Sciences, Yancheng 224002, China; sdauxxx@163.com; 3School of Agriculture and Environment, The University of Western Australia, Crawley, WA 6009, Australia

**Keywords:** soil profile, biochar, soil fertility, nutrient transport, nitrogen

## Abstract

To investigate the responses of crop production and soil profile nutrient status to biochar (BC) application, we conducted a soil column experiment considering two BC addition rates (0.5 and 1.5 wt% of the weight of 0–20 cm topsoil) combined with two nitrogen (N) input levels (low N: 144 kg ha^−1^, LN; high N: 240 kg ha^−1^, HN). The results showed that BC application increased the soil pH. The soil pH of the 0–10 cm profile under LN and the 20–40 cm profile under HN were both significantly increased by 0.1–0.2 units after BC addition. Under LN, BC addition significantly increased NH_4_^+^-N (17.8–46.9%), total N (15.4–38.4%), and soil organic carbon (19.9–24.0%) in the 0–10 cm profile, but decreased NH_4_^+^-N in the 20–30 cm soil profile and NO_3_^−^-N in the 10–30 cm profile by 13.8–28.5% and 13.0–34.9%, respectively. BC had an increasing effect on the available phosphorus, the contents of which in the 10–20 and 30–40 cm soil profiles under LN and 20–30 cm profile under HN were significantly elevated by 14.1%, 24.0%, and 23.27%, respectively. However, BC exerted no effect on the available potassium in the soil profile. BC had a strong improving effect (15.3%) on the wheat yield, especially the N144 + BC0.5% treatment, which could be compared to the HN treatment, but there was no yield-increasing effect when high N fertilizer was supplied. In summary, BC improved the fertility of agriculture soil (0–20 cm) with wheat. In particular, low N inputs together with an appropriate rate of BC (0.5 wt%) could not only achieve the low inputs but also the high outputs in wheat production. In future study, we will compare the effects of multiple doses of N and BC on soil fertility and crop production.

## 1. Introduction

Nitrogen (N) is a key nutrient that limits crop growth and is essential for improving crop yield and quality [1]. In order to pursue economic benefits, farmers have excessively applied chemical N fertilizers in the production process, resulting in crop yields and N utilization efficiency (NUE) no longer increasing continuously. Consequently, large N losses including nitrate (NO_3_^−^-N) leaching and ammonia and nitrous oxide emissions threaten the quality of the water and the atmospheric environment [2,3,4,5]. Therefore, coupling N with other soil additives to ensure food security while reducing the environmental loss of N fertilizer is a major ongoing concern in global agricultural production and in the ecological environment.

In recent years, biochar (BC) has gained popularity due to its potential in soil improvement and environmental remediation [6,7]. BC is a porous carbon material that is produced by the pyrolysis of organic materials (e.g., agricultural residues) under low-oxygen but high-temperature conditions, and it is characterized by a high carbon stability, a large specific surface area, and a strong adsorption capacity [8]. Ample evidence has shown that BC addition could increase crop NUE and yield and improve soil fertility [9,10].

Assessing the response of crop yields to N fertilizer application and exogenous substance addition is a key research area for assessing the effectiveness of farm management practices. However, the most recently published studies have assessed the effectiveness of BC application in farmland based on only a single N application rate [11]. In addition, studies have shown that crop yield is not positively correlated with BC application. For example, Zhang et al. [12] found that BC addition had an enhancing effect on tobacco yield when BC was applied at 0.2–1.0% of the tillage soil’s weight, while it inhibited its growth when applied at a 5% ratio. Meanwhile, Uzoma et al. [13] found that maize yields were higher in treatments with a BC application of 12 t hm^−2^ than in the 20 t hm^−2^ treatment. Therefore, in view of the resource properties of N fertilizers and BC, evaluating the crop yield increase effect of BC under different N supply conditions and its relationship with BC dosage is helpful to consolidate the optimal combination of BC and N fertilizer.

Many studies have been conducted to demonstrate the effect of BC in increasing and/or retaining soil nutrients. Han et al. [14] showed that BC significantly increased the soil organic matter (SOM), NO_3_^−^-N, and total N in a 0–20 cm soil layer. Moreover, Muhammad et al. [15] demonstrated that BC significantly increased the soil’s organic carbon (SOC) and the soil’s available N and phosphorus (P) contents in a 0–15 cm soil layer. However, most of these studies focused on the effects of BC on the soil properties in the tilled layer (above 20 cm) and neglected the longitudinal effects on the soil profiles. Heymann et al. [16] found that BC particles are further decomposed into micro and nanoparticles in the soil environment and vertically transported to a deeper soil layer; meanwhile, Xue et al. [17] suggested that there is a difference between the nature of granular BC that undergoes migration and that of normal BC. This will affect the subsoil’s properties. For example, Xia et al. [18] found that BC significantly enhanced the SOM contents and the ammonium (NH_4_^+^-N) contents in a 20–40 cm soil layer. Wang et al. [19] found that BC significantly increased the SOC content in the 140–160 cm range but that the microbial mass carbon contents decreased slightly. In addition, many scholars have argued that the distribution of nutrients in the soil profile can also be used as an indicator of soil fertility, characterizing the sustainable use of soil to a certain extent [20,21,22]. Therefore, it is necessary to evaluate the response of soil nutrients to BC from the vertical perspective of the soil profile.

In summary, this study took the interactions of BC and N fertilizer on the soil profile’s nutrients as the entry point and conducted an indoor soil column test to explore the effects of different BC application levels on the spatial longitudinal variation in soil nutrients and the responses of crop yield and NUE to BC under different N supply levels. This research can provide a scientific basis for what is a reasonable application amount of BC and N fertilizer.

## 2. Results

### 2.1. Soil Properties

#### 2.1.1. pH

The pH changes in the soil profile are shown in Figure 1. When BC addition was not considered, the pH showed a pattern of increasing first and then decreasing with the soil profiles at a low N supply, while it showed a pattern of decreasing first and then increasing with a high N supply. BC increased the soil pH, especially at a high N application rate. As shown in Figure 1a, the effect of BC application on pH improvement under a low N level was mainly observed in the surface soil, and the pH of the N144 + BC1.5% treatment in the 0–10 cm profile was significantly higher than that of N144 + BC0.5% and N144 by 0.11 and 0.20 units, respectively. In the 10–20 cm profile, the N144 + BC1.5% treatment’s pH was significantly higher than that of N144 by 0.04 units. In contrast, the increase in pH thanks to BC at a high N level was observed in the deeper soils (Figure 1b), where the pH in the N240 + BC1.5% and N240 + BC0.5% treatments was significantly higher than that in the N240 treatment by 0.20 and 0.10 units, respectively, in the 20–30 cm profile. And the pH in the N240 + BC1.5% and N240 + BC0.5% treatment was significantly higher than in the N240 treatment by 0.14 and 0.10 units in the 30–40 cm profile.

#### 2.1.2. NH_4_^+^-N, NO_3_^−^-N, and Total N

The changes in the NH_4_^+^-N contents are shown in Figure 2. After BC was applied, the NH_4_^+^-N contents showed a trend of first decreasing and then increasing at both N supply levels (gradually decreasing at a depth of 0–30 cm and increasing at a depth of 30–40 cm). BC had a significant effect on the increase in the NH_4_^+^-N contents in the 0–10 cm profile. The NH_4_^+^-N contents in the N144 + BC0.5% and N144 + BC1.5% treatments were significantly higher than the content in the N144 treatment by 37.0% and 46.9%, respectively. For the 20–30 cm profile, the NH_4_^+^-N contents in N144 + BC0.5% and N144 + BC1.5% were significantly lower than the content in N144 by 28.5% and 13.8%, respectively (Figure 2a).

BC addition resulted in a decrease in the soil NO_3_*^−^*-N, and this tendency was more significant under high BC application amounts but low N levels (Figure 3). The soil NO_3_*^−^*-N was significantly reduced by 22.4% and 22.0% in the N144 + BC0.5% and N144 + BC1.5% treatments, respectively, compared with the levels observed in the N144 treatment. In the 20–30 cm profile, the soil NO_3_*^−^*-N content in the N144 + BC1.5% treatment was significantly lower (by 34.3%) than that in the N144 treatment (Figure 3a).

BC addition significantly increased the total N content in the surface soil layer, and the total N contents increased with the BC application rate (Figure 4). In the 0–10 cm soil profile, the contents of total N were significantly higher in the N144 + BC1.5% treatment than in N144, but there was no significant difference between N144 and N144 + BC0.5%. The total N contents were significantly higher in N240 + BC0.5% and N240 + BC1.5% than in N240 (Figure 4b).

#### 2.1.3. SOC

The SOC contents were higher at a high N level than at a low N level, and the SOC contents at both N supply levels increased with increasing BC application rates after its addition (Figure 5). At the low N level, the SOC contents decreased significantly in the 10–20 cm profile but changed steadily in the 20–30 and 30–40 cm profile (Figure 5a). In the 0–10 cm profile, the SOC contents reached their maximum in N144 + BC1.5%, which was significantly higher than the 19.9% and 21.1% contents achieved in the N144 and N144 + BC0.5% treatments, respectively. However, in the 10–20 cm profile, the SOC content in the N144 + BC1.5% was significantly 12.2% higher than that in N144. At a high N level, the SOC contents were positively correlated with the BC application rate. The SOC contents under all treatments decreased continuously at depths of 10–20 and 20–30 cm but increased at 30–40 cm. In addition, the SOC contents of all soil profiles reached their maximum in the N240 + BC1.5% treatment, and the SOC contents were 24.3%, 4.6%, 20.6%, and 12.6% higher than the content measured in the soil without BC, with soil depth arranged from that closest to the top of the soil to the bottom, respectively (Figure 5b).

#### 2.1.4. Soil-Available P (AP) and Potassium (AK)

The results of our study showed that, when BC application was not taken into account, the differences in the soil’s AP contents at different N application levels were small and showed a pattern of first decreasing and then increasing with soil depth, reaching a maximum at the 30–40 cm profile (Figure 6). At a low N level, BC mainly affected the AP contents in the 10–20 cm profile rather than the 30–40 cm soil profiles (Figure 6a). The AP contents in the 10–20 cm profile under the N144 + BC0.5% and N144 + BC1.5% treatments were significantly increased by 24.0% and 21.9%, respectively, compared with that at the same depth with the N144 treatment; while, in the 30–40 cm profile, the AP contents under the N144 + BC0.5% and N240 + BC1.5% treatments were significantly increased by 16.6% and 12.1%, respectively, compared to the N144 treatment. Meanwhile, at a high N supply level, an effect of applying BC on the AP content was observed in the 20–30 cm profile (Figure 6b), where the N240 + BC0.5% treatment had the greatest AP content, which was significantly higher by 23.3% and 17.4% than the contents of the soil treated with N240 and N240 + BC0.5%, respectively.

Figure 7 shows that the AK contents first decreased in the 10–20 cm profile and then increased in the 30–40 cm profile under both N supply levels when the BC applied was not considered. Under a low N level, the effect of BC on the AK content was remarkable in the 0–10 and 30–40 cm profiles (Figure 7a). The AK contents in the 0–10 cm profile reached its maximum in the N144 + BC0.5% treatment, which was 12.0% significantly higher than that in the N144 + BC1.5% treatment. However, the difference was not significant between the treatments without BC application. In the 30–40 cm profile, the AK contents were significantly higher under N144 + BC0.5% than N144 and N144 + BC1.5% by 5.7% and 4.5%, respectively. Under a high N supplementation, BC had a significant effect only in the 30–40 cm profile (Figure 7b). N240 had the highest AK content, which was 12.9% and 16.2% significantly higher than that of N240 + BC0.5% and N240 + BC1.5%, respectively.

### 2.2. Wheat Yield and the Related Agronomic Traits

As shown in Table 1, the wheat yield was higher in the treatments with a high N than in those with a low N content. At a low N level, BC application significantly reduced the number of grains but increased the weight of a thousand grains, so it still increased the wheat yield. The maximum yield was recorded in the N144 + BC0.5% treatment, which was 15.3% significantly higher than the yield in N144. The N144 + BC1.5% treatment also tended to increase the yield of wheat, but the difference between it and N144 was not significant. The effect of BC on the yield components and the yield of wheat was not so remarkable at a high N level, with the maximum yield being recorded under the N240 + BC1.5% treatment, but it was not significantly different from the rest of the treatments at the same N application level.

As it can be seen from Table 2, the N recovery efficiency (NRE) and the NUE were lower at a high N level than at a low N level, with a reduction of 5.9–15.5%. At the same N fertilizer application rate, the NRE and the NUE showed a trend of first decreasing and then increasing with the increase in the BC application rate, which indicated that N the fertilizer with a low BC rate has a risk of increasing N loss, but the possibility of N loss decreased with the increase in the BC application rate, and this effect was more pronounced under the low N level.

## 3. Discussion

### 3.1. Soil Properties

The results showed that there was a positive correlation between the pH and the BC in the same soil profile, indicating a positive effect of BC on ameliorating soil acidification, which is similar to the results reported by Chintala et al. [23]. Soil pH is governed by salt-based ions, and BC contains a large number of alkaline ash elements, and these components are all soluble [24], which can increase the salt-based saturation of acidic soils, which, in turn, can exchange hydrogen ions and exchangeable aluminum in the soil and improve soil acidification. In addition, our study also found that the enhancement of the pH by the addition of BC varied under different levels of N application, and the effect of BC on the soil pH at the low N level was concentrated in the tillage layer (0–20 cm), while it tended to be weaker in the deep layer (20–40 cm). However, the enhancement of pH at the high N level was mainly observed in the deep layer (Figure 1b). This may be due to the fact that, when N fertilizers are applied in larger quantities, part of the downward leaching of N will carry BC with it to the deeper soil layers, thus changing the pH of the deeper soils. On the other hand, it may be that, under a level of high N content, the decomposition rate of BC into particles may be accelerated, and its vertical transport rate to deeper soil would also be accelerated.

Soil NH_4_^+^-N and NO_3_^−^-N, as mineralized N in the soil, are the main forms of N absorbed by plants, accounting for about 70% of the total amount of anions and cations absorbed [25]. Many studies have pointed out that the problem of N loss in agricultural production systems is relatively serious, which may lead to the deterioration of groundwater quality and the loss of biodiversity [26,27]. Our study showed that the NH_4_^+^-N contents of tillage soil increased with the increase in BC application. Previous results by Ahmad et al. [28] showed that the N mineralization rate in soil was enhanced by BC application, which explained the increase in NH_4_^+^-N contents due to BC addition in the present experiment. In addition, the results of our experiment showed that, in the 0–10 cm soil profile, the NH_4_^+^-N and NO_3_^−^-N contents under the treatment with added BC were higher, while, at the deeper soil profile, the NH_4_^+^-N and NO_3_^−^-N contents under the BC treatment were lower than those measured under the N fertilizer-only treatment, and this pattern was even more significant at the low N level. This suggests that BC application under a moderate N application rate contributed to an increase in the retention capacity of the soil’s N, weakened the rate of the transport of NH_4_^+^-N and NO_3_^−^-N from the upper part of the soil downwards, and reduced the loss of leaching N, which was also consistent with the results of Zhang [29] and Kameyama [30]. The reason behind these phenomena is that BC has a huge specific surface area and a surface functional group, which can strongly adsorb the dissolved NH_4_^+^-N and NO_3_^−^-N in the soil and make them stay in the surface soil [31].

In this study, the effect of N application on the vertical distribution of the total N content in the soil profile was small when BC addition was not considered. However, BC application significantly increased the total N contents of the 0–10 cm surface soil. The total N contents were significantly and positively correlated with the BC application rate, which was basically consistent with the results of Dong et al. [32]. As previous work, N contained in BC significantly enhanced the soil’s N pool [33]. It may be one of the reasons explaining the increase in the total N contents. In addition, Chen et al. [34] reported that BC increased microbial N demand, stimulated soil N cycling (including nitrification and denitrification processes), and enhanced microbial N immobilization in soil, further explaining the BC-induced increase in N supplementation from a microscopic perspective. Zhang et al. [35] also pointed out that a high application rate of BC can lead to N being retained in the soil and capture more N, a phenomenon which was also consistent with our results.

In this experiment, BC addition significantly increased the SOC contents of the tillage soil, which is consistent with the findings reported by Kimetua et al. [36]. The first reason for this is that BC contains a large amount of elemental C, which forms difficult-to-decompose SOC after being applied to the soil and is not easily changed in the short term [37]. Secondly, the surface passivation of BC interacts with the soil to form a protective matrix, which increases the oxidative stability of organic carbon and contributes to the accumulation of SOC [38]. In addition, BC has a huge specific surface area, and its adsorption of small organic molecules catalyzes the formation of new organic matter on its surface [39]. Under high N application conditions, BC application also elevated the SOC content in deeper soils (Figure 5b), which further suggested that BC undergoes vertical migration downward under high N application.

The results of this study showed that BC addition played a positive role in increasing the AP and AK contents in the surface soil at the low N application level. Laird et al. [40] found that the pore structure of BC delayed the release of fertilizer nutrients and was able to reduce nutrient losses and increase soil P and K effectiveness, and this result was confirmed by the present study. In contrast, the AP contents of the treatments with high BC application were lower in the 0–10 cm profile at the high N application level, i.e., a high amount of BC was not favorable for the AP content. It is as if the effect of too much BC in the soil was only dominated by the adsorption of soil P and, thus, led only to small increase in the AP contents [41]. Moreover, the soil AK contents increased with increasing BC application rates in our study. Consistently, Zhang et al. [42] found that applying BC at 0%, 0.5%, 1.0%, and 2.0% rates significantly increased the AK contents in red and yellow-brown soils. Our study also found that, at a low N level, the effects of BC on AP and AK were mainly observed in the topsoil layer, while, at a high N level, the effects were mainly manifested in the deep soil layer (Figure 6b). This effect indicates that BC was more likely to migrate to the deep soil layer at the high N level.

### 3.2. Wheat Yield and NUE

The application of N fertilizer is a key measure to improve wheat yield, and, with the increase in N fertilizer, wheat yield tends to increase [43], a matter which was confirmed by our results. Data from Table 1 indicate that, at the low N level, the treatments with BC application at a 0.5% rate had higher grain yields than those at a 1.5% rate, though the difference was not significant. When the fertilizer N rate was low, high soil C/N levels limited the mineralization of the soil’s organic N, which reduced the efficacy of the N fertilizer applied to the soil [44]. The current experimental results showed that the NH_4_^+^-N and NO_3_^−^-N contents in the soil under the N144 + BC0.5% treatment were higher than those under the N144 + BC1.5% treatment. In this experiment, the NRE and NUE of wheat at the high N level were significantly lower than those at the low N level, which was consistent with the results of Zhao et al. [45]. According to their report, the excessive application of N fertilizer led to a large amount of N loss. It has been shown that BC can significantly increase crop uptake of N fertilizer [46]. However, in the results of our experiment, the effect of the increase in BC content on the NUE was not significant, and the addition of BC even showed a trend of decreasing the NUE (Table 2). This may be due to the high carbon content of BC, which can result in excessive soluble SOC content and the consequent competition between soil microorganisms and wheat for the soil’s N [47]. Moreover, BC itself is characterized by a large specific surface area and high adsorption, which adsorbs the N in soil. In addition, Dong [48] showed a significant increase in ammonia emissions of 16.8–47.7% after BC application during the wheat season, which was the results of enhanced soil respiration after BC addition. Overall, from the perspective of economic benefit and environmental friendliness, we recommend treatment with N144 + BC0.5% for practices in the agricultural field.

## 4. Materials and Methods

### 4.1. Experiment Setup

#### 4.1.1. Experimental Site and Soil and Biochar Properties

The soil column simulation experiment was carried out in the research greenhouse on the top floor of the College of Forestry, at Nanjing Forestry University, which was only equipped with a roof (made of fiberglass-reinforced polyester), not affecting the lighting and ventilation conditions, which were similar to the field conditions. The tested soil was collected from paddy fields in Zhoutie town, Yixing city, Jiangsu province (31°07′ N, 119°31′ E). The study area has an average annual temperature and rainfall of 15.7 °C and 1108 mm, respectively. In May 2021, we collected texted soils from a 15 ha paddy field in the order of 0–20, 20–40, and 40–60 cm profiles, respectively. The paddy soil samples were mixed according to the profile, air-dried for about 10 days, sieved through a 2 mm sieve, and repacked into soil columns at a bulk density similar to the site conditions. The soil columns were made of PVC material with a height of 60 cm, a diameter of 30 cm, and a closed bottom. The soil at different profile levels was repacked into the soil column according to the sampling orders, tamped, and irrigated with deionized water to settle [49]. The basic properties of the topsoil (0–20 cm) were as follows: a pH (1:2.5 soil–water ratio) of 6.36; a cation exchange capacity of 19.6 cM kg^−1^; SOM 2.28%; available N 0.38 g kg^−1^; and the total N, total P, and total K contents were 1.56 g kg^−1^, 0.96 g kg^−1^, and 4.12 g kg^−1^, respectively. The BC raw material used in the experiment was wheat straw, which was made into tested biochar at a cracking temperature of 500 °C [50], with a pH of 9.8, a total N content of 8.1 g kg^−1^, a SOC content of 37.5%, and a specific surface area of 32 m^2^ g^−1^.

#### 4.1.2. Experimental Design and Management

The levels of N fertilizer application were 144 and 240 kg ha^−1^ (N144 and N240), and the amounts of BC applied were 0.5% and 1.5% (BC0.5% and BC1.5%; percentage of the weight of the 0–20 cm topsoil), respectively. A control treatment (CK) without fertilizer N addition was also set up to calculate the NUE. Therefore, there were seven treatments in this study: i.e., CK; N144; N144 + BC0.5%; N144 + BC1.5%; N240; N240 + BC0.5%; and N240 + BC1.5%. Each treatment was replicated three times; therefore, there were 21 soil columns in total for the experiment.

Before the sowing of the wheat, BC was applied to the 0–20 cm surface soil layer of the corresponding experimental treatments, and then we manually mixed the soil and BC evenly. The wheat (Var. Ningmai 13, provided by the Jiangsu Academy of Agricultural Sciences) was sown on 10 November 2021. Fifty seeds were sown in each column (pot). Basal N, P, and K fertilizers were applied at the same time as sowing. Seedling emergence was counted one month after sowing, and the wheat seedlings were thinned to 30 per pot for harvest.

The experiment was conducted in a conventional water and fertilizer management mode according to local farmers. Fertilizer N was applied three times as a basal fertilizer (30%) on November 10, as a tiller fertilizer (30%) on 21 January in 2022, and as a spike fertilizer (40%) on the 18 March 2022, respectively. And P (90 kg P_2_O_5_ hm^−2^) and K (120 kg K_2_O hm^−2^) fertilizers were applied as basal fertilizers one time. The N, P, and K fertilizers were provided by urea (46% N), calcium superphosphate (12% P_2_O_5_), and potassium chloride (60% K_2_O), respectively.

### 4.2. Sample Collection and Measurement

#### 4.2.1. Soil Sampling

After harvesting on the 24 May 2022, three sampling points were selected from each soil column, and soil profile samples were collected at 0–10, 10–20, 20–30, and 30–40 cm depths using a soil auger (50 mm in inner diameter). Part of the fresh soil samples, measuring 2 mm, was screened to determine the NH_4_^+^-N and NO_3_^−^-N contents. The remaining soil samples were air-dried to determine the pH and the total N, SOC, AP, and AK contents. The basic physicochemical properties of the soil were determined according to the conventional methods of soil agrochemical analysis [51]. The details are as follows: the NH_4_^+^-N content was determined by the indophenol blue colorimetric method; the NO_3_^−^-N contents were determined by the phenol sulfonic acid colorimetric method; the pH was determined by the potentiometric method; the total N contents were determined by Kjeldahl nitrogen determination; the SOC contents were determined by the oxidation of potassium dichromate-external heating method; and the AP and AK contents were determined by 0.5 M dm^−3^ NaHCO_3_.

#### 4.2.2. Wheat Yield and NUE

Wheat is harvested at physiological maturity to determine the grain yield and its components. The agronomic traits related to the yield (plant height, panicle number, and 1000-grain weight) were recorded. Twelve wheat plants were randomly selected and divided into straw and seed parts. They were oven-dried at 105 °C for 30 min and then dried until they reached a constant weight at 75 °C, weighed, ground into powder using a high-speed crusher (Yongguangming Medical Instrument C., Ltd., FW-100, Beijing, China), and passed through a 2 mm nylon screen. The N contents in the grains and in the straws were measured by the Kjeldahl method after H_2_SO_4_-H_2_O_2_ wet digestion [52].
Apparent recovery of N fertilizer (NRE, %) = (N uptake by plants with N application − N uptake by plants without N application)/Nitrogen application × 100%(1)
N uptake efficiency (NUE, kg/kg) = N uptake by plant/N application rate(2)

### 4.3. Statistical Analysis

The data were analyzed using SPSS 26.0 (SPSS Institute Inc., Chicago, IL, USA) for one-way ANOVA and LSD tests, and different lowercase letters indicate significant differences between treatments at the significance level of *p* < 0.05.

## 5. Conclusions

In our study, through soil column simulation experiments, the following conclusions were drawn: Firstly, the application of BC could improve soil acidification and increased the nutrient contents of NH_4_^+^-N, NO_3_^−^-N, total N, SOC, AP, and AK in the 0–20 cm soil profile at both N supply levels. Secondly, BC has two effects on the vertical distribution of nutrients in the soil profile: (1) BC addition could effectively sequester N in the topsoil and reduce the risk of its leaching to the deeper soil layers, and the N sequestration capacity was closely related to the rate of BC applied; (2) Partially aged BC may move downward with water leaching, thus affecting nutrient content in deep soils, which is more obvious at high N levels. The N fertilizer level was still the dominant factor in increasing the wheat yield. BC application exerted little effect on seed N uptake and NUE but could increase the wheat yield. Overall, the yield increase effect of BC on wheat was better under a low N rate. Nevertheless, this experiment was a short-term soil column experiment. Hence, the long-term effects of BC on soil profile nutrients and crop yield need to be further studied, especially at the field scale.

## Figures and Tables

**Figure 1 plants-13-00614-f001:**
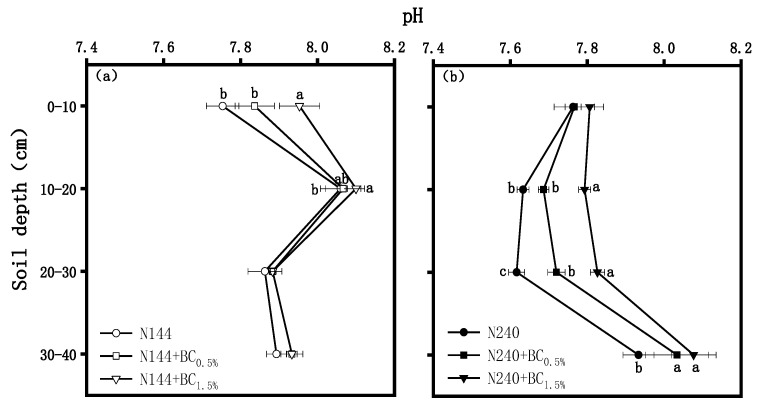
Effects of biochar (BC) applied at two rates (0.5 and 1.5 wt% of the weight of the top 20 cm of soil) on the pH in a soil profile under different nitrogen (N) supplies, i.e., (**a**) LN, 144 hm^−2^ and (**b**) HN, 240 kg hm^−2^. The values are means ± SD (*n* = 3), and different letters at the same soil depth indicate statistically significant differences at the 0.05 level.

**Figure 2 plants-13-00614-f002:**
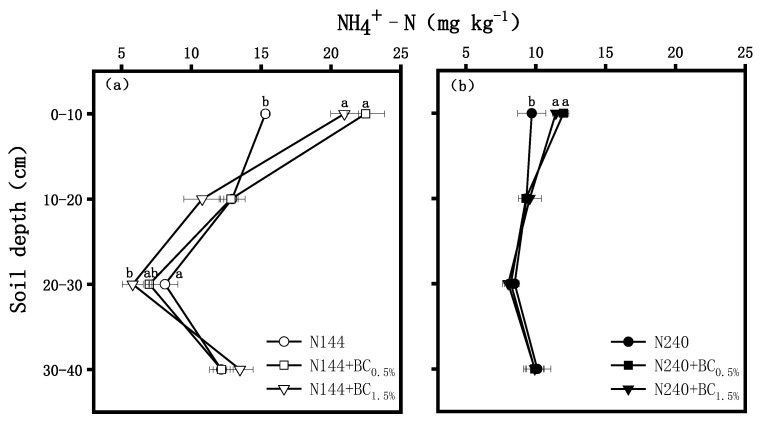
Effects of biochar (BC) applied at two rates (0.5 and 1.5 wt% of the weight of the top 20 cm of soil) on contents of ammonium (NH_4_^+^-N) in a soil profile under different nitrogen (N) supplies, i.e., (**a**) LN, 144 hm^−2^ and (**b**) HN, 240 kg hm^−2^. The values are means ± SD (*n* = 3), and different letters at the same soil depth indicate statistically significant differences at the 0.05 level.

**Figure 3 plants-13-00614-f003:**
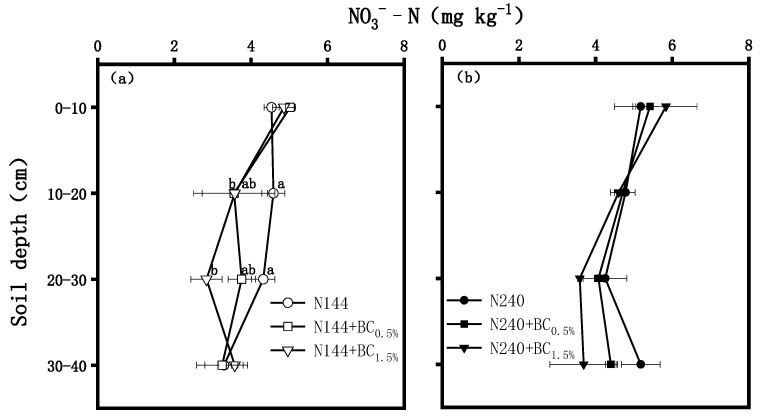
Effects of biochar (BC) applied at two rates (0.5 and 1.5 wt% of the weight of the top 20 cm of soil) on the contents of nitrate (NO_3_^−^-N) in a soil profile under different nitrogen (N) supplies, i.e., (**a**) LN, 144 hm^−2^ and (**b**) HN, 240 kg hm^−2^. The values are means ± SD (*n* = 3), and different letters at the same soil depth indicate statistically significant differences at the 0.05 level.

**Figure 4 plants-13-00614-f004:**
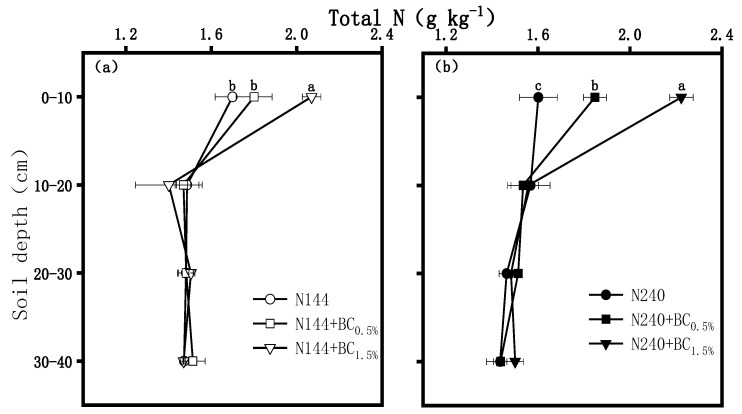
Effects of biochar (BC) applied at two rates (0.5 and 1.5 wt% of the weight of the top 20 cm of soil) on the contents of the soil’s total nitrogen (N) in a soil profile under different nitrogen supplies, i.e., (**a**) LN, 144 hm^−2^ and (**b**) HN, 240 kg hm^−2^. The values are means ± SD (*n* = 3), and different letters at the same soil depth indicate statistically significant differences at the 0.05 level.

**Figure 5 plants-13-00614-f005:**
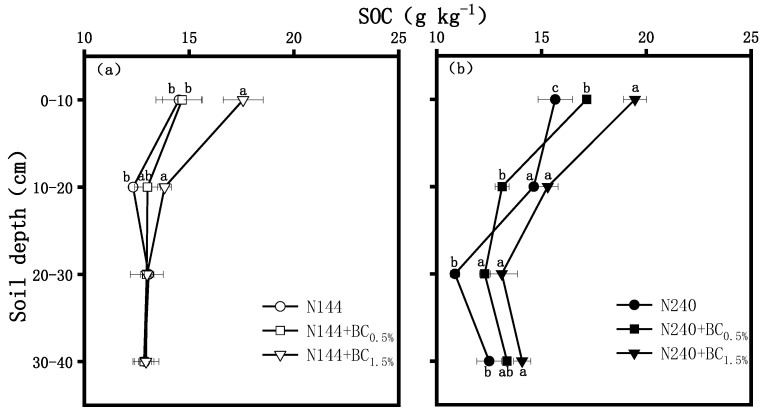
Effects of biochar (BC) applied at two rates (0.5 and 1.5 wt% of the weight of the top 20 cm of soil) on the contents of soil organic carbon (SOC) in a soil profile under different nitrogen (N) supplies, i.e., (**a**) LN, 144 hm^−2^ and (**b**) HN, 240 kg hm^−2^. The values are means ± SD (*n* = 3), and different letters at the same soil depth indicate statistically significant differences at the 0.05 level.

**Figure 6 plants-13-00614-f006:**
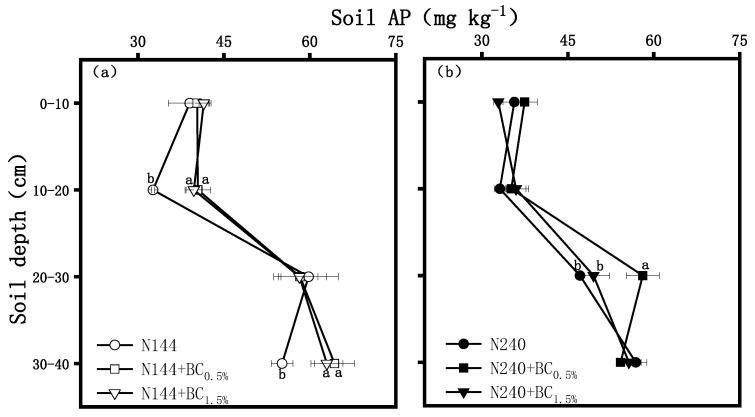
Effects of biochar (BC) applied at two rates (0.5 and 1.5 wt% of the weight of the top 20 cm of soil) on the contents of soil-available phosphorus (AP) in a soil profile under different nitrogen supplies, i.e., (**a**) LN, 144 hm^−2^ and (**b**) HN, 240 kg hm^−2^. The values are means ± SD (*n* = 3), and different letters at the same soil depth indicate statistically significant differences at the 0.05 level.

**Figure 7 plants-13-00614-f007:**
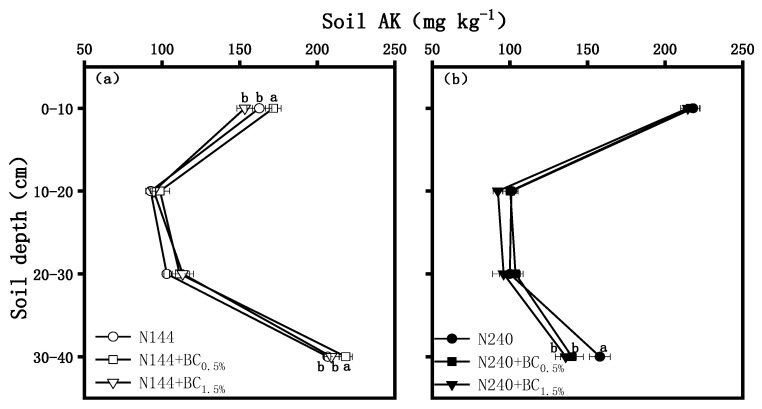
Effects of biochar (BC) applied at two rates (0.5 and 1.5 wt% of the weight of the top 20 cm of soil) on the contents of soil-available potassium (AK) in a soil profile under different nitrogen supplies, i.e., (**a**) LN, 144 hm^−2^ and (**b**) HN, 240 kg hm^−2^. The values are means ± SD (*n* = 3), and different letters at the same soil depth indicate statistically significant differences at the 0.05 level.

**Table 1 plants-13-00614-t001:** Responses of wheat grain yield and the related agronomic traits to biochar (BC) addition at two rates (0.5 and 1.5 wt% of the weight of the top 20 cm of soil).

Treatment	Grain Number per Pot	12-Average Plant Height (cm)	1000 Grains’ Weigh (g)	Theoretical Yield (g pot^−1^)
CK	142.33 ± 25.14 d	45.4 ± 3.89 c	67.58 ± 7.73 d	284.33 ± 80.26 e
N144	433.33 ± 35.35 a	69.2 ± 3.41 ab	110.04 ± 16.6 c	1561.26 ± 442.95 d
N144 + BC0.5%	376 ± 17.25 bc	70.1 ± 3.54 ab	123.25 ± 6.36 bc	1843.43 ± 209.13 abc
N144 + BC1.5%	328.67 ± 9.97 c	63.7 ± 3.67 ab	141.91 ± 12.73 ab	1751.87 ± 301.77 cd
N240	410 ± 30.09 ab	68.7 ± 3.92 ab	131.48 ± 4.84 ab	2498.99 ± 427.63 ab
N240 + BC0.5%	414 ± 15.72 ab	65.6 ± 2.76 ab	134.33 ± 2.51 ab	2393.37 ± 180.23 abc
N240 + BC1.5%	370.67 ± 13.03 bc	71.6 ± 7.87 a	147.81 ± 5.85 a	2636.92 ± 357.70 a

Note: Values are means ± SD (*n* = 3), and different letters in the same column indicate statistically significant differences at the 0.05 level.

**Table 2 plants-13-00614-t002:** Responses of wheat nitrogen (N) uptake, recovery efficiency (NRE), and use efficiency (NUE) to biochar (BC) addition at two rates (0.5 and 1.5 wt% of the weight of the top 20 cm of soil).

Treatment	Grain N Uptake (g pot^−1^)	Straw N Uptake (g pot^−1^)	Total N Uptake (g pot^−1^)	NRE (%)	NUE (kg kg^−1^)
CK	0.16 ± 0.04 e	0.05 ± 0.02 d	0.21 ± 0.03 d	-	-
N144	0.83 ± 0.09 cd	0.26 ± 0.03 bc	1.08 ± 0.11 bc	58.8 ± 7.25 a	0.74 ± 7.25 ab
N144 + BC0.5%	0.81 ± 0.15 d	0.22 ± 0.01 c	1.03 ± 0.15 c	55.7 ± 10.36 ab	0.70 ± 10.36 abc
N144 + BC1.5%	0.85 ± 0.12 bcd	0.24 ± 0.01 ab	1.09 ± 0.11 bc	59.8 ± 7.36 a	0.75 ± 7.36 a
N240	1.13 ± 0.13 a	0.30 ± 0.06 ab	1.43 ± 0.18 a	49.8 ± 7.47 ab	0.59 ± 7.47 bcd
N240 + BC0.5%	1.07 ± 0.06 ab	0.23 ± 0.01 c	1.30 ± 0.07 ab	44.3 ± 2.99 b	0.53 ± 2.99 d
N240 + BC1.5%	1.05 ± 0.03 abc	0.34 ± 0.02 a	1.39 ± 0.02 a	48 ± 0.77 ab	0.57 ± 0.77 cd

Note: Values are means ± SD (*n* = 3), and different letters in the same column indicate statistically significant differences at the 0.05 level.

## Data Availability

Data are available upon request.

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
