# Peer review of "Interaction of Biochar Addition and Nitrogen Fertilizers on Wheat Production and Nutrient Status in Soil Profiles"

_plants, 2024, doi:10.3390/plants13050614_

Round 1

Reviewer 1 Report

Comments and Suggestions for Authors

Review Report

Manuscript ID: plants-2848018

Title: Interaction of biochar addition and nitrogen fertilizer input on production of wheat and nutrient status of soil profiles

Authors: Jiale Liu, Zirui Chen, Si Wu, Haijun Sun, Jincheng Xing, Zhenhua Zhang

Dear Authors,

Please provide detailed, step-by-step answers to all my comments.

I see a few details that need to be improved or corrected. I described all these details in the reviewer's comments in the pdf file.

General comments:

Please use the suggested by MDPI notation: area hectare (hm2); yield (Mg hm-2); g kg-1; mg kg-1;  g lysimeter-1; cation exchangeable capacity (cM kg-1); specific area (m2 g-1) throughout the whole manuscript.

Please carefully check the spaces, double spaces, and no spaces in the whole manuscript body.

Detailed comments:

Title

Please consider changing “Interaction of biochar addition and nitrogen fertilizer input on production of wheat and nutrient status of soil profiles”

to

„Interaction of Biochar Addition and Nitrogen Fertilizers on Wheat Production and Nutrient Status in Soil Profiles”

Authors

Line 11 Please delete doubled Correspondence: Correspondence

Abstract

Line 13: Please consider changing “soil column” to “lysimeter” I think lysimeter is a much more appropriate term than soil column.

Keywords

Line 29: Please consider changing “biomass charcoal” to “biochar”

1.    Introduction

Line 56: t/ha → Mg hm-2

Line 62, 64, 71: soil → soil layer

Line 79: “soil column” → “lysimeter”

2. Results

2.1.2. NH4+-N, NO3–-N, and total N

Line 124: Figure 3: please do not use a slash (/) mg/kg mg kg-1

Line 132, 149: Figure 4, 5: please do not use a slash (/) g/kg g kg-1

Line 177, 181: Figure 6, 7: please do not use a slash (/) mg/kg mg kg-1

Line 194, Table 1: What does grain number mean? (Number of grains per? per plant, per spike, per lysimeter, per surface area of lysimeter? The same question considers the Theoretical yield

Line 203, Table 2: Did you use pots? I think lysimeter is a much more appropriate term than pot.

Material and Methods

4.1. Experiment Setup 309

4.1.1. Experimental site and soil properties  Experimental site and soil properties and physiochemical properties of biochar

Line 311: Rain shelter? Difficult to understand. Please try to clearly explain. Was the experiment carried out on the roof (roof, terrace, patio, deck)?

Line 313: lighting and ventilation conditions close to the field  similar to the field

Lines 325-331: Please see general comments

Line 342: interplanted thinned

Line 354: Why did the authors write that used an auger with an inner diameter of 5 cm to remove stones if the soil (before the experiment started) was sieved by 2 mm sieve?

Line 364: mol/L M dm-3

Line 370: crushed crushed with cutting mill (producer, city, country)

Line 371: decoction wet mineralization

4.3. Statistical analysis

SPSS 26.0 software (IBM Inc. Amok, NY, USA)

5. Conclusions

OK

Author Contributions: OK

Funding: OK

Data Availability Statement: OK

Conflicts of Interest: OK

References OK

Dear Authors,

After taking into account the corrections, I recommend your manuscript be accepted for publication in Plants MDPI Journal.

Yours sincerely

Reviewer

Author Response

See it in the attached file.

Reviewer 2 Report

Comments and Suggestions for Authors

The present study, which attempts to observe the effect at different depths in the soil, taking as variables the amount of N applied and the effect of biochar dose at the microscale, is interesting. However, I believe that the paper has important shortcomings that may make its publication difficult. Firstly, the experimental design is certainly questionable, the work is justified in extending the conditions of the different doses studied, but only two doses of N and two of biochar were studied. There are many issues not well described in the Materials and Methods that should be clarified. Controls where only biochar is added are not included. There is also a lack of an adequate multifactorial statistical study that includes all factors and examines their interaction. At the level of results, there are contradictions and inconclusive results that are difficult to explain, e.g. that increasing the dose of N does not increase crop yield. The conclusions do not reflect the results obtained. The background should be improved. In short, I consider the publication of this manuscript difficult in view of the results obtained. At the discretion of the editor regarding its acceptance, the results should be very well justified and contrasted with a greater bibliographic background. At the writing level, it needs a thorough revision, also in English. More details can be found in the attached file.

Comments on the Quality of English Language

Writing and English should be reviewed extensively.

Author Response

See them in the attached file.

Reviewer 3 Report

Comments and Suggestions for Authors

The submitted manuscript to PLANTS-MDPI entitled “Interaction of biochar addition and nitrogen fertilizer input on production of wheat and nutrient status of soil profiles” is interesting to investigate. BUT, following are the comments that need to be addressed:

The future perspectives are missing in abstract.

According to authors, the pH and soil phosphorus have positive correlation, however in my opinion (based on literature), If the pH is low (acidifying nature), the P availability can be increased for plants which can be taken up by the plants. How do you respond to this constraint?

It is strongly suggested that the results should be written in fold changes or percentage changes.

Line 378: replace *applied* with *application*

How did the authors choose to study specific amounts of nitrogen and biochar? Were there any preliminary experiments to verify these amounts?

What was the rationale of choosing these soil depths to study? Were there any previous studies to verify with?

Author Response

See them in the attached file.

Round 2

Reviewer 2 Report

Comments and Suggestions for Authors

First of all, I would like to thank the authors for clarifying some of my questions. I think that the manuscript remains basically the same, as no changes have been made at the level of presentation of results, statistics, nor significantly at the background level. I think that the experimental design could be improved as I indicated. However, I understand that substantial improvements have been made to the manuscript that partially or fully justify the issues raised and point out some problems in the research. Therefore, despite the identified deficiencies, I consider that it could be accepted for publication at the editor's discretion, taking into account the sufficiency of the justifications provided.

Comments on the Quality of English Language

In my opinion, minor editing of English language is required in this 2nd version.